# Enhancing Sustainability in PLA Membrane Preparation through the Use of Biobased Solvents

**DOI:** 10.3390/polym16142024

**Published:** 2024-07-16

**Authors:** Giovanna Gomez d’Ayala, Tiziana Marino, Yêda Medeiros Bastos de Almeida, Anna Raffaela de Matos Costa, Larissa Bezerra da Silva, Pietro Argurio, Paola Laurienzo

**Affiliations:** 1Institute of Polymers, Composites and Biomaterials, National Research Council (IPCB-CNR), Via Campi Flegrei, 34, 80078 Pozzuoli, NA, Italy; giovanna.gomez@ipcb.cnr.it (G.G.d.); paola.laurienzo@ipcb.cnr.it (P.L.); 2Department of Chemical Engineering, Federal University of Pernambuco, Recife 50740-520, PE, Brazil; yeda.oliveira@ufpe.br; 3CICECO–Aveiro Institute of Materials, Department of Chemistry, University of Aveiro, 3810-193 Aveiro, Portugal; anna.raffaela@ua.pt; 4Postgraduate Program in Materials Science and Engineering, Federal University of Rio Grande do Norte, Natal 59078-970, RN, Brazil; larissabezerramat@gmail.com; 5Department of Environmental Engineering, DIAm, University of Calabria, Via Pietro Bucci CUBO 44/A, 87036 Rende, CS, Italy; pietro.argurio@unical.it

**Keywords:** polylactic acid, biopolymer, biobased solvents, NIPS, green membranes

## Abstract

For the first time, ultrafiltration (UF) green membranes were prepared through a sustainable route by using PLA as a biopolymer and dihydrolevoclucosenone, whose trade name is Cyrene™ (Cyr), dimethyl isosorbide (DMI), and ethyl lactate (EL) as biobased solvents. The influence of physical-chemical properties of the solvent on the final membrane morphology and performance was evaluated. The variation of polymer concentration in the casting solution, as well as the presence of Pluronic^®^ (Plu) as a pore former agent, were assessed as well. The obtained results highlighted that the final morphology of a membrane was strictly connected with the interplaying of thermodynamic factors as well as kinetic ones, primarily dope solution viscosity. The pore size of the resulting PLA membranes ranged from 0.02 to 0.09 μm. Membrane thickness and porosity varied in the range of 0.090–0.133 mm of 75–87%, respectively, and DMI led to the most porous membranes. The addition of Plu to the casting solution showed a beneficial effect on the membrane contact angle, allowing the formation of hydrophilic membranes (contact angle < 90°), and promoted the increase of pore size as well as the reduction of membrane crystallinity. PLA membranes were tested for pure water permeability (10–390 L/m^2^ h bar).

## 1. Introduction

Membrane separation technologies are utilized in various sectors, including the food industry and water treatment, due to their numerous advantages over traditional separation methods. These technologies enable highly selective separations using equipment that is easy to operate, requires low maintenance, and consumes minimal energy, aligning well with environmentally friendly standards [1,2,3,4]. Among them, ultrafiltration (UF) is considered one of the most promising separation methodologies because of its cost-effectiveness, simplicity of equipment layout, and high efficiency [5,6], and it is widely employed in different applicative fields, as water treatment, biotechnology, and food processing [7,8,9]. Commercial UF membranes are generally based on synthetic nonbiodegradable polymers, including polyethersulfone (PES) and polyvinylidene fluoride (PVDF) [1,10,11,12], but, in spite of their great efficiency, their use is not convenient, since they cannot be disposed of and, therefore, they contribute to the crucial issue of plastic waste accumulation [13]. In recent decades, there has been a growing interest in minimizing the environmental impact associated with the use of these membranes by making the entire production chain sustainable. This involves using biodegradable polymers made from renewable source materials and processing them into eco-friendly and fully recyclable bioplastics [1].

In this context, the use of polylactic acid (PLA) is particularly promising, since it can be produced from renewable biomasses and recycled after usage through composting methodologies; in addition, it exhibits very low toxicity [14,15,16,17]. Moreover, the great potential of this biopolymer lies in its ability to be processed through different techniques, such as thermoforming, blow molding, extrusion via casting, and injection molding [18], and in its commercial availability (large-scale production) in a wide range of grades [19].

Recently, PLA has been investigated for synthesizing advanced, biodegradable, and sustainable membranes [20,21,22], either as the main matrix or as a modifier agent for the improvement of specific features of membranes [14,23,24].

Porous PLA membranes can be manufactured through different procedures, including electrospinning, particle leaching, gas foaming, and phase separation [25,26]. Among them, NIPS is one of the most widely employed due to its many advantages, such as versatility, scalability, simplicity, and ability in room temperature processing [20]. This process involves the phase separation of a polymer from a homogeneous dope solution by adding an appropriate nonsolvent, which induces a phase separation into a polymer-rich phase, responsible for the membrane structure, and a polymer-poor phase, which gives rise to the pore formation [27]. Using this methodology, various PLA membranes were developed, demonstrating particularly promising potential for treating wastewater with high nutrient content [15].

The choice of solvent, in addition to the type of polymer, is crucial to make the entire membrane production process truly sustainable. Nowadays, N,N-dimethylformamide (DMF), N,N-dimethylacetamide (DMA), chloroform, and 1,4-dioxane are among the most widely used solvents for the manufacturing of PLA-based membranes [15,28], but, unfortunately, their use is particularly hazardous for environment and human health, due to their high toxicity [28].

Minbu et al. [29] reported the preparation of PLA membranes via NIPS by using 1,4-dioxane as a solvent and different surfactants as pore former agents. The obtained membranes were proposed as suitable for microfiltration (MF) applications. Xing et al. [30] prepared PLA membranes with dioxane as a solvent and tested different coagulation baths. The resulting membranes, having a pore size in the MF range (~9 μm), showed a sponge-like structure with a high crystallinity when ethanol was used in a mixture with water as a precipitation bath. Nassar et al. [15] described the preparation of asymmetric PLA membranes for wastewater treatment, using PVP as a pore former agent and dimethyl acetamide (DMAc) as a solvent, through NIPS varying the polymer concentration in the casting solution. Eco-friendly and biodegradable membranes, composed of PLA, hydroxyapatite as filler, and polydopamine as surface coating, were developed for ultrafiltration applications by Ouda et al. [31]. N-Methyl-2-pyrrolidone (NMP) was used as solvent.

Although the cited literature examples present promising results for advancements in the polymeric membrane preparation via phase inversion by using PLA as polymeric materials, the reported procedures involved the use of very toxic and environmentally polluting solvents. For this reason, in recent years, there has been a great interest in identifying possible nontoxic and environmentally benign solvents that could limit the use of traditional ones, reducing human health risks and environmental damage through biomass valorization [1,29,32,33,34,35].

In the last years, biorenewable Cyr and DMI, as well as EL, have been investigated as potential candidates for the replacement of toxic solvents.

Cyr is a green dipolar aprotic solvent prepared by pyrolysis of cellulose-containing biomass, and it was recognized with the “Bio-Based Chemical Innovation of the Year” award in 2017 [36]. The increasing interest in Cyr is due to its similarity with traditionally used organic aprotic solvents, such as NMP and DMA, in terms of polarity, density, and miscibility with water [37]. Although it exhibits similar polarity to these solvents, it is safer to handle, environmentally friendly, and highly stable against oxidation. Upon degradation, it converts into water and carbon dioxide. Cyr is particularly promising for membrane preparation because it is high-boiling, fully miscible with water, and can be easily removed.

EL is an environmentally benign solvent with effectiveness comparable to petroleum-based solvents. It has attracted a lot of attention in recent years, as it is obtained by the esterification reaction of ethanol and lactic acid, which can be generated from biomass raw materials through fermentation [38]. It is generally recognized as safe, and its low viscosity, poor volatility, broad liquid temperature range, and high specific solvent abilities make it particularly promising for several industrial applications. Recently, ethyl lactate solvent has received attention for dissolving PLA because it is easily recyclable, completely biodegradable, nonozone depleting, and nontoxic, and consequently, through its use, it is possible to obtain fully green and sustainable PLA-based membrane [39].

Dimethyl isosorbide (DMI), a nontoxic, water-soluble, high-boiling (boiling point: 93–95 °C at 0.10 mmHg; 235–237 °C at 760 mmHg) green solvent derived from D-sorbitol, is considered one of the top-10 biobased platform chemicals. Due to its renewable starting materials, DMI is a suitable substitute of the more toxic currently used solvents, such as dimethylsulfoxide (DMSO), N,N-dimethylformamide (DMF), and DMAc [40,41].

The aim of this study was to prepare porous PLA membranes through NIPS, using the above-mentioned biobased solvents, and to characterize them in terms of morphology, porosity, pore size, wettability, and mechanical properties. It is well known that solvent characteristics, such as viscosity, dielectric constant, polarity, and boiling point, are crucial, since they strongly affect the morphology and performance of resulting membranes. In light of this, the produced membranes were accurately characterized, in terms of morphology, porosity, pore size, wettability, and tensile properties, in order to assess the effect of the employed solvent on their characteristics and to identify the optimal alternative to the traditional solvents for the manufacture of PLA-based membranes for ultrafiltration applications.

## 2. Experimental Section

### 2.1. Materials

LUMINY^®^ LX175 transparent PLA resin, which has a high viscosity and low flow and is amorphous, was supplied by TotalEnergies Corbion (Gorinchem, The Netherlands). The polymer was stored in a vacuum oven at a temperature of 40 °C before its use. Pluronic^®^ F-127 powder was purchased from Sigma-Aldrich (Schnelldorf, Germany). The three nontoxic solvents employed for preparing the casting solution, i.e., Cyr (dihydrolevoglucosenone, BioRenewable, ≥98.5% purity), DMI (1,4:3,6-dianhydrosorbitol 2,5-dimethyl ether, BioRenewable, ReagentPlus^®^, ≥99.0% purity), and EL (Ethyl 2-hydroxy propionate, natural, ≥98% purity), were all were purchased from Sigma-Aldrich (Schnelldorf, Germany). Ultrapure water was used as a nonsolvent.

### 2.2. Membrane Preparation

Porous PLA-based membranes were prepared according to the NIPS method [40,41]. Homogeneous PLA solutions were prepared using Cyr, DMI, and EL as polymer-dissolving solvents. PLA pellets were dried at 40 °C for 24 h under vacuum and were added to the different solvents to get final concentrations of 8 and 10% (*w*/*w*). In addition, Plu was employed as a porogen agent to improve the porosity, and solutions of PLA 8% and Plu 2% were also prepared in all the chosen solvents.

The polymer dissolution was performed under stirring for 24 h at different temperatures, depending on the employed solvent: 85 °C for Cyr and DMI and 100 °C for EL. Once the polymer was completely dissolved, the resulting solutions were degassed and cast on a glass plate by means of a casting knife with a set thickness of 250 μm and immediately immersed in an H_2_O coagulating bath at room temperature. In particular, Cyr and DMI solutions were cooled to room temperature before casting, whereas EL solutions were cast while still warm (70 °C). After coagulation, the obtained membranes were thoroughly washed with water and dried under vacuum at RT.

### 2.3. Thickness

The thickness of membranes was measured using a digital micrometer (IP65 Mitutoyo, Kawasaki, Japan) and the resulting data were reported as the averaged value of 20 different measurements for each membrane.

### 2.4. Porosity

Membrane porosity was determined according to the procedure described by Sukitpaneenit et al. [42]. Three different samples of each dry membrane were weighed and soaked in kerosene liquid for 24 h. Membranes did not swell in the selected solvent, and their porosity was calculated according to the following Equation (1):(1)$$Porosity\;\left(\%\right)=\frac{\frac{W_{w}-W_{d}}{\rho_{w}}}{\frac{W_{w}-W_{d}}{\rho_{w}}+\frac{W_{d}}{\rho_{p}}}\times 100$$
where *W_w_* and *W_d_* are the mass of solvent (kerosene)-impregnated and dry membranes, respectively, *ρ_w_* is the kerosene density (0.82 g/cm^3^), and *ρ_p_* is the polymer density. In this approach of porosity calculation, it was assumed that the entire void volume was filled with kerosene.

### 2.5. Pore Size Distribution

Average pore size and pore size distribution within the membranes were determined through a liquid-gas displacement technique, employing Porewick^®^ (Sigma-Aldrich, Milan, Italy, superficial tension 16 dyne/cm) as the wetting agent, with a surface tension of 16 dyne/cm. This experimental procedure was facilitated through the utilization of a capillary flow porometer apparatus, specifically the CFP-1500 AEXL model manufactured by Porous Materials Inc., Ithaca, NY, USA. The instrument meticulously recorded the spectrum of membrane pore sizes ranging from the minimum to the maximum, with particular emphasis on the mean flow pore diameter as indicative of the central pore size parameter.

Furthermore, the elucidation of the pressure requisite for the displacement of the wetting liquid from the pores was pursued in accordance with a defined mathematical Equation (2) [43], thereby providing a quantitative understanding of the fluid dynamics within the porous structure.
p = γ cos θ [dS/dV]pore(2)

In the provided equation, p represents the difference in gas pressure acting across the wetting liquid within the pore, while γ indicates the surface tension of the wetting liquid. The contact angle of the liquid on the surface of the pore is denoted by θ. dV refers to the volume of liquid displaced by the gas within the pore, and dS corresponds to the increase in solid/gas surface area resulting from the displacement of the liquid. The expression dS/dV is directly linked to the pore size. In instances where the cross-section of the pore is noncircular, the pore size is characterized in terms of an equivalent pore diameter (D), which is established through the following relationships (Equations (3) and (4)):[dS/dV] pore = [dS/dV] cylindrical pore(3)
[dS/dV] cylindrical pore = 4/D(4)

Expressed in terms of pore diameter (D), Equation (1) can be simplified as demonstrated in Equation (5), which states that the gas pressure (p) across the wetting liquid within the pore is inversely proportional to the pore diameter (D), with the constant 4 multiplied by the surface tension of the wetting liquid (γ) and the cosine of the contact angle (θ).
p = 4 γ cos θ/D(5)

When employing a low surface tension wetting liquid such as Porewick^®^, the equation is further simplified (Equation (6)), as the contact angle for low surface tension liquids tends toward zero.
p = 4 γ/D(6)

For cases where the pore conformation lacks regularity, as observed in etched stainless-steel discs, cylindrical track etched membranes, and woven fabrics, the capillary flow porometer was configured with a default tortuosity factor of 0.715. The maximum pore size diameter (d) was calculated using Equation (7), where d represents the maximum pore diameter expressed in micrometers, and C is a constant (0.0286 bar).
d = C γ/p(7)

The mean flow pore diameter was determined from Equation (7), utilizing the pressure (p) at which the wet flow was equivalent to half the dry flow. Pore size distribution was derived using Equation (8), where D denotes the pore size distribution, Q signifies the filter flow percentage, and L represents the preceding value.
D = (*Q* − *Q*L)/dL − *d*(8)

Prior to pore size analysis, membranes were submerged in Porewick^®^ for 24 h. Nitrogen gas was employed, gradually introduced with increasing pressure into the membrane over time. Measurements of both gas pressure and permeation flow rates across the dried membrane were obtained, facilitating the calculation of the final pore size distribution. All measurements were performed in triplicate.

### 2.6. Morphological Analysis (SEM)

Membrane morphology was investigated through scanning electron microscopy (SEM) using a Quanta 200 FEG (FEI, Hillsboro, OR, USA) microscope. Lower, upper, and cryogenically fractured cross-sectional surfaces were observed. Samples were coated with a 20 nm thick gold/palladium layer with a modular high-vacuum coating system, Emitech K575× (Emitech, Montigny, France).

### 2.7. Thermal Characterization (DSC)

Thermal properties of PLA membranes were assessed using a Q2000 T zero differential scanning calorimeter (DSC) by TA Instrument (New Castle, DE, USA), equipped with a liquid nitrogen accessory for fast cooling. Samples of approximately 7 mg were equilibrated at −20 °C and then heated to 250 °C at a rate of 10 °C/min. All the measurements were carried out under a nitrogen flow rate of 30 mL/min. The glass transition temperature (T_g_) was evaluated by applying the first derivative procedure. The degree of crystallinity (*X_c_*) for each membrane was calculated according to the following Equation (9) [44]:(9)$$X_{c}=\frac{{\Delta H}_{m}-{\Delta H}_{cc}}{{\Delta H}_{100\%}^{0}}\times 100$$
where ${\Delta H}_{m}$ and ${\Delta H}_{cc}$ are melting and cold crystallization enthalpy, respectively, and ${\Delta H}_{100\%}^{0}$ is the melting enthalpy of a 100% crystalline PLA (93 J/g) [45,46].

### 2.8. Surface Wettability

Surface wettability of membranes was assessed using a MicroDrop^®^ (First Ten Angstroms Inc., Milan, Italy) contact angle meter with a high-speed framing camera by determining the contact angle (θ) through the drop shape analysis. In particular, 4 μL of distilled water was dropped on a flat membrane surface at room temperature by using a syringe. The contact angle (defined as the angle between the selected surface and the tangent drawn from the edge of the drop) was measured using FTA1000 Manual Drop Shape Analysis Software version 2.0 (FTA Inc., Portsmouth, VA, USA). The reported results were averaged over 4 measurements for each sample.

### 2.9. Mechanical Tests

The tensile properties of PLA membranes were determined using an Instron 4505 instrument. The analyses were performed on dumbbell-shaped film specimens (type IV according to ASTM D638), at a constant deformation rate of 5 mm/min. Young’s modulus, elongation at break (ε_b_), and strain at break (σ_b_) were measured and the reported results were averaged over 10 measurements for each membrane.

### 2.10. Pure Water Permeability (PWP)

A laboratory crossflow cell manufactured by DeltaE srl (Milan, Italy) was utilized to conduct permeate water permeability (PWP) experiments at a temperature of 25 °C. The experimental setup involved directing pure water through a membrane with an area of 21 cm^2^, by means of a peristaltic pump (VELP Scientifica SP311, Usmate Velate, Italy). Prior to initiating the permeability tests, a 30 min equilibration period was observed, during which the transmembrane pressure was maintained at 0.1 bar. In addition, 1.2/1.0/0.8 bar as transmembrane pressures, disconnected by a stabilization period of one to another of 20 min, were applied and the permeate was collected in 60 s. PWP was calculated by applying the following Equation (10):PWP = Q/A t p,(10)
where A is the membrane area (expressed in m^2^); p is the pressure (expressed in bar); Q is the permeate volume in liters; and t is the time (expressed in hours).

## 3. Results and Discussion

The use of biobased solvents in the production of polymer-based membranes by NIPS offers a sustainable alternative, reducing reliance on fossil fuels and mitigating environmental impact. This approach adheres to green chemistry principles, fostering eco-friendly separation processes crucial for a greener future. The development of PLA membranes using the three novel biobased solvents and their thorough characterization may represent a significant stride toward these objectives.

Table 1 shows the casting solution composition, which was varied in terms of solvent (Cyr, DMI, and EL), polymer concentration (8 and 10%), and presence of 2% Plu, and also the viscosity measured for each dope solution.

In Table 2, some of the most relevant physical-chemical properties of Cyr, DMI, and EL are reported. It is worth noting that DMI exhibits the highest molecular weight and boiling point, while EL the lowest. However, Cyr is characterized by the highest density and a complete miscibility with water.

### 3.1. Morphology

The polymer-solvent distance is crucial in determining membrane morphology and performance. It influences various characteristics, such as pore size, structure, and overall efficiency, making it a critical factor in optimizing the membrane fabrication process. Specifically, a close match between the solubility parameters of the polymer and the solvent can result in improved membrane formation and performance. In fact, along with the kinetic factors, Hansen parameters play significant roles in the formation of polymer membranes and the development of their unique structures, including the formation of finger-like/spongy structures. The Hansen solubility parameters can be used to predict the compatibility or miscibility between a selected polymer and the solvent, providing insight into the distance between them in a solubility space. The interaction distance of a polymer is a way to gauge how easy or difficult it is to dissolve the polymer and can be calculated using the Euclidean distance formula reported by Hansen [49]. If the distance is small, it indicates that the solvents used need to have a very similar distribution of cohesive forces in order to effectively dissolve the polymer. In other words, a smaller difference in Hansen parameters between the polymer and solvent indicates a higher likelihood of miscibility, leading to the formation of a homogeneous and stable solution. These solubility parameters include three components: δd, which represents the London dispersion forces; δp, the polar or dipole-dipole forces; and δh, representing the hydrogen bonding forces. Comparing Cyr, DMI, and EL solubility parameters and the related PLA–solvent distance, it is worth observing that the lowest distance was observed for Cyr (1.16 Mpa^0.5^), whereas DMI exhibited an intermediate value (3.33 Mpa^0.5^) between Cyr and EL, with the latter showing the highest value of 7.37 Mpa^0.5^. The ester functional groups (-COO-) on the PLA backbone favor the interaction with the polar groups of Cyr, promoting enhanced solvation of PLA chains and therefore making the formation of PLA/Cyr mixtures thermodynamically favored. This evidence was confirmed during membrane preparation by visually monitoring the time required for the nascent film to detach from the glass support. When Cyr was used, this process took 15 min, indicating a very low solvent–nonsolvent exchange rate. The high polymer-solvent affinity was further confirmed by the stability of the solutions, even at room temperature. On the contrary, EL gave rise to unstable solutions that were not thermodynamically favored, as indicated by the high PLA–solvent distance and the higher temperature required to dissolve the polymer (100 °C) compared to that used for Cyr and DMI (85 °C). Hence, from a thermodynamic point of view, solvent–nonsolvent exchange happened faster for the EL/water system than Cyr/water because of the lower affinity between PLA and EL. While DMI has moderate polarity due to its hydroxyl groups (-OH) and ether linkages, which can form hydrogen bonds with PLA ester groups, its interaction with PLA may not be as strong as that with Cyr. As a result, although DMI can dissolve some PLA, polymer–solvent interactions may exhibit weaker intermolecular forces compared to PLA–Cyr interactions, and it may not be as effective or efficient as Cyr Indeed, DMI presents an intermediate solvent–polymer distance compared to Cyr and EL, but still considered suitable for a PLA solubilization, since a distance less than 4 indicates good affinity between the polymer and the solvent [50]. The final structure of a membrane is often the result of complex interplays between thermodynamic and kinetic factors, and primarily viscosity, which can affect the kinetics of processes involved in membrane formation.

For all of the used solvents, the viscosity of PLA solutions increased as a function of concentration, following this order: 10 ˃ 8-2 ˃ 8. The interactions established between PLA and solvent molecules influenced the viscosity of polymer solutions. An increase in polymer concentration determined an increase in solution viscosity through enhanced polymer entanglement, and vice versa. Plu additive, having a high molecular weight (12,500 g/mol), contributed to an increase in the solution viscosity due to the higher resistance of molecules to flow. Among the three solvents, Cyr yielded solutions with higher viscosity compared to both DMI and EL. As reported by Kol et al. [51] from a thermodynamic point of view, when a good solvent is used, the polymer segments prefer to be surrounded by solvent molecules rather than by other segments, while in poor solvents, the polymer molecules try to minimize the area of contact with the solvent molecules. In fact, in a poor solvent, like EL, PLA chains exhibited a low attraction to the solvent and tended to adhere to segments of nearby polymer molecules. On the contrary, in a favorable solvent, such as DMI, or, even more, Cyr, solvent–PLA contacts were preferred, discouraging direct interchain interactions. Consequently, in a good solvent, a polymer molecule experienced greater ease in moving and rotating freely among its neighboring molecules than in a poor solvent, where freedom was restricted, potentially resulting in the aggregation of polymer molecules. On the basis of these considerations, it was expected that Cyr, being the most thermodynamically favored solvent among the three selected solvents, should lead to solutions with low viscosity. However, due to the very high intrinsic dynamic viscosity of 14.5 cP [52] at 25 °C (for comparison, DMI: 6.80 cP and EL: 4.7 cP), the PLA-Cyr solutions, also visually, were more viscous in comparison to the analog DMI and EL-based solutions (Table 1).

SEM images, reported in Figure 1, revealed peculiar behavior for the PLA membrane formation starting from Cyr, which was likely due to the high viscosity of the solvent on one hand, and the strong polymer-solvent and solvent–nonsolvent affinity on the other hand. Polymer precipitation may be influenced by the full miscibility of Cyr and water, which may result in an instantaneous demixing process, but also by the high viscosity of the dope solution, which may promote a delayed phase inversion process. PLA membranes prepared with this solvent exhibited an irregular, asymmetrical morphology in which fingerlike macrovoids are surrounded in a continuous PLA matrix. A cellular structure was observed between the macrovoids, and the wall structure surrounding these voids is highly compact. The obtained membrane morphology was similar for all three membranes prepared with Cyr, and the dense matrix is mainly ascribable to the very high viscosity of the polymeric dope [53]. In fact, even when Plu was used as a hydrophilic pore former agent, the membrane morphology did not present substantial differences in comparison to the other two membrane types (10Cyr and 8Cyr). In accordance with Ren and Wang [54], the growth of macrovoids is an instantaneous phenomenon beneath the top layer (liquid-liquid demixing), while the polymer concentration did not change significantly in the area distant from the interface of the membrane solution. Hence, the phase inversion and the related formation of the cellular structure is a delayed process within the matrix beneath the surface layer (crystallization) induced by the coarsening of the polymer-lean phase, at the late stage of phase separation.

When DMI was used as a solvent, membrane morphology consisted of large and digitate macrovoids. The preparation of membranes with finger-like morphology has attracted great attention from researchers due to the high permeability of the obtained membranes [55]. The obtained membrane structure was the result of the trade-off between kinetic effect (polymeric solution viscosity) and thermodynamic instability. For DMI-based membranes, the thermodynamic instability seemed to be the dominant factor, since the phase separation occurs faster, resulting in a finger-like structure. The presence of Plu as an additive provoked an increase in polymeric system instability, leading to a faster exchange rate of the solvent and nonsolvent. This led to large macrovoids over the whole cross-section of the prepared membrane. By using EL, the polymer precipitation at the nonsolvent–polymer interface occurred very fast due to the low affinity between the solvent and PLA and, at the same time, the low viscosity of the dope solution. The high demixing rate led to the formation of a dense skin top layer, which prevented the solvent from entering the forming polymer matrix. The presence of a dense top layer, reducing the water intake rate, favored the polymer crystallization process over liquid–liquid demixing, promoting the formation of granular polymeric spheres. The granular structure was more accentuated at higher polymer concentrations (PLA 10%) and the viscosity increased as the polymer particles, beneath the dense surface layer, had time to crystallize.

### 3.2. Thickness, Porosity and Pore Size

Table 3 reports the thickness and porosity of the developed PLA membranes. Comparing the casting solution with the same composition, thickness decreased in the following order: PLA-Cyr ˃ PLA-DMI ˃ PLA-EL.

The obtained data revealed that the thickness was strictly related to the membrane morphology. In fact, when Cyr was used, the membranes were thicker than the membranes obtained with DMI and EL, especially in comparison to EL-based ones, which promoted the formation of the thinnest membranes (Table 3). For each solvent, the dope solution composition strictly influenced membrane thickness, whose maximum value corresponds to the highest polymer concentration (10%), followed by an intermediate value relative to the PLA-Plu system, down to the lowest value, corresponding to PLA 8%. The membrane porosity reduced as the polymer concentration increased for all of the tested solvents. Plu is a block copolymer consisting of 30% PPO blocks and 70% PEO blocks, is formed by hydrophobic polypropylene oxide (PPO) segments in the middle and hydrophilic PEO segments at both ends and, due to its amphiphilic character, conferred a noticeable hydrophilicity to the nascent films, acting as pore former agent [56]. However, with its high molecular weight, (M_w_ = 12,600 g mol^−1^) and the increase in dope solution viscosity, it could represent an obstacle for nonsolvent inflow in the polymeric matrix. As reported by Plisko et al. [57], block copolymers tend to form micelles by self-assembly in polymeric solutions due to the different affinities of solvents depending on the different blocks of copolymer [57,58,59]. With the immersion of the cast dope solution in the coagulation bath, a double phase inversion occurred: on one hand, the solvent nonsolvent exchange, and on the other, Plu self-assembly. The combination of the two processes promoted an enhanced membrane porosity with respect to that observed without the additive.

It is well known that a UF membrane with a narrow and uniform pore size distribution is resistant to getting clogged, and it is very useful for treating wastewater. Membranes with many small, evenly sized pores can filter water more effectively. The pore size of the obtained prepared membranes lies in the range of ultrafiltration (Figure 2).

Figure 3 shows the pore size distribution of PLA membranes. When Cyr was used as a solvent, the resulting membranes exhibited a very sharp unimodal pore size distribution with a mean flow pore diameter of 0.2–0.3 μm. In the case of DMI, the membrane prepared starting from PLA8-2 showed the highest detected pore size (0.09 μm), with a 35% pore number and an average pore dimension between 0.1 and 0.15 μm. 8DMI membrane displayed a uniform narrow particle size distribution and a 70% pore number with a 0.04 μm dimension. It has often been reported in the literature [60] that an increase in the dope solution viscosity related to an increase in polymer content leads to a reduction in the pore size with a narrower distribution and a concomitant increase in porosity. Accordingly, both 8 and 10EL exhibited an average pore size of 0.05 μm and 0.03 μm, respectively, and a slight increase of up to 0.06 μm when Plu was added to the casting dope (Figure 3).

The pore distribution for EL membranes was in wider ranges than those of the analog membranes prepared with Cyr and DMI, probably due to the significantly lower viscosity. Moreover, as reported by Aijaz et al. [61], the presence of Plu increased the hydrophilic surface groups, promoting the concentration of the block copolymer. The abundance of hydrophilic surface moieties in the nascent film enables water molecules diffusion through its pores, thus ensuring wetting of the membrane’s underside despite the inherent water-repellent characteristics of PLA. Plu is able to improve UF membrane performance in terms of resistance to fouling, making the membrane more effective in filtering out unwanted substances. Moreover, it exhibits both hydrophilic and hydrophobic parts, making it compatible with polymeric material and ensuring long-term stability. The water-attracting segments enhance the membrane’s affinity for water, while the water-repelling segments facilitate the copolymer integration into the PLA membrane structure. A notable characteristic of Plu copolymer is its ability to spontaneously arrange itself, influencing the membrane’s formation process. This self-assembly capability can be harnessed to produce uniform pores within the membrane, thereby improving its filtration performance.

### 3.3. Contact Angle and Mechanical Properties

The contact angle measurements allow us to determine the hydrophilic or hydrophobic membrane characteristics. In general, if a water droplet on the membrane surface has a contact angle of less than 90°, the material is considered hydrophilic. Otherwise, it is hydrophobic. The contact angles of the PLA membrane are shown in Table 4.

It is well recognized that naturally, PLA is a hydrophobic material (contact angle > 90°). Table 4 confirmed that, by increasing the polymer concentration, regardless of the solvent, the contact angle increased due to the hydrophobic nature of the material [15]. However, the contact angle is also strictly related to other parameters, including dimension and distribution of surface pores, as well as membrane surface morphology [62]. Varying the dope solution composition, pore geometry, and porosity also changed. This is clearly detectable by using Plu as a pore former agent in the casting solution. In accordance with Aisueni et al. [63], for the membranes that evidenced more dead-end pores than interconnected ones, the hydrophilic character was more pronounced due to the higher water adsorption of liquid. This was particularly evident for the 8-2Cyr membranes, which, together with the evident presence of dead-end pores on the surface, also combined an increase in porosity. Also, Zhang et al. [64] reported that membranes with small pore sizes with uniform distribution and a high porosity were characterized by a lower hydrophilicity due to the presence of more voids on the surface. The addition of Plu had a positive effect on membrane hydrophilicity. In fact, in accordance with Plisko et al. [57], the observed decrease in water contact angle related to the use of Plu can be ascribed to the migration of amphiphilic chains to the surface of the selective layer and formation of hydrophilic nanodomains that cover the surface and pore walls of the membrane selective layer. In fact, hydrophilic PEO blocks of Plu can form aggregates [56] on the polymer-water interfacial surface during membrane formation due to low surface tension, while hydrophobic PPO blocks of Plu are compactly bound with polymer PLA chains.

Significant differences were observed in the mechanical performance of the membrane depending on the solvent employed for fabrication, as shown in Table 4. Typically, membranes crafted with Cyr demonstrated augmented Young’s modulus in contrast to those prepared with DMI. Membranes produced with EL exhibited a similar trend to those obtained with DMI, showing elevated Young’s modulus juxtaposed with diminished elongation at break. The addition of Plu 2%, as represented by 8-2EL, appeared to marginally decrease both Young’s modulus and elongation at break, similar to what was observed with Cyr. Of particular note, EL-derived membranes showed the highest Young’s modulus among all solvents tested, except for 8-2Cyr, concurrently exhibiting the lowest elongation at break. These nuanced findings provide invaluable insights into tailoring the mechanical properties of PLA-based membranes, thereby facilitating the optimization of fabrication processes as a function of the specific applications.

In addition, Cyr-derived membranes manifested superior elongation at break in comparison to their DMI counterparts. Particularly noteworthy is 8-2Cyr, exhibiting the highest Young’s modulus and elongation at break amongst all specimens tested, indicative of the potential efficacy of integrating 2% Plu to enhance mechanical properties. During phase inversion, as the solvent and nonsolvent slowly separated, pores were formed within the polymer matrix. The slower demixing rate in the case of Cyr–water systems enabled the formation of smaller and more uniformly distributed pores, which contributed to improved mechanical properties such as elongation at break and resistance to deformation. These results can be explained by considering the roles of the solvents and the addition of Plu in the casting solution, along with the variations in polymer content. The rate at which the solvent exchanges with the nonsolvent during membrane formation can influence the formation of the membrane structure and the alignment of polymer chains. A slow exchange rate allows for a gradual transition from a homogeneous solution to a polymer-rich phase, facilitating the alignment of polymer chains in the membrane matrix. Cyr may be flushed out from the polymer matrix more slowly compared to DMI and EL, allowing for better alignment and packing of PLA molecules, thus enhancing mechanical properties. Well-aligned polymer chains result in a more ordered and compact membrane structure, leading to increased mechanical strength and stiffness. During phase inversion, the slow demixing rate in the case of Cyr–water systems enabled the formation of smaller and more uniformly distributed pores (Figure 3), which contributed to improved mechanical properties such as enhanced elongation at break and better resistance to deformation in comparison to the other two solvent–polymer systems. Plu molecules can act as template agents during membrane formation, influencing the arrangement of polymer chains and the formation of pores, which in turn affects the mechanical behavior of the membranes. Higher polymer content can result in membranes with higher stiffness (Young’s modulus) but lower elongation at break due to the denser packing of polymer chains.

### 3.4. Thermal Properties

Figure 4 shows the DSC curves of the different PLA membranes, recorded during the first heating run to evaluate their intrinsic thermal properties.

For all of the samples, the thermograms showed similar endo- and exothermic transitions. In particular, a glass transition around 60 °C (T_g_) associated with a large endothermic peak relative to the phenomenon of enthalpy relaxation was detected. In general, this process is related to the release of stresses accumulated during the manufacturing and storage of materials and occurs when samples are heated beyond their T_g_. For NIPS membranes, the stress is likely imparted by the rapid coagulation of polymer chains in the nonsolvent, which forces them into an unrelaxed conformation [28]. A cold crystallization exothermic peak between 80 and 110 °C appears in all of the DSC curves, suggesting that PLA was in a quasi-amorphous state in the obtained membranes, followed by the melting endotherm at a temperature around 145 °C. The melting endotherm appears shifted into two peaks, which correspond to the α’- and α-crystals, typical of PLA [64].

All of the thermal parameters of the prepared PLA membranes are listed in Table 5.

In the case of Cyr, the increase in PLA concentration determined an increase in T_g_ of membranes, probably due to the larger extent of intermolecular interactions and entanglements between polymer chains, which make the membrane more rigid, whereas no considerable variations in T_cc_ and T_m_ were detected. Moreover, an increase in crystallinity degree was observed with the increase in polymer concentration. In general, at low concentrations, the poor mobility of polymer chains at room temperature makes the contact between adjacent chains and their rearrangement in a crystalline structure difficult. On the other hand, when the concentration increases, the entanglement between molecular chains is facilitated and, therefore, the crystallinity is enhanced [65]. The addition of Plu caused a considerable reduction in the degree of crystallinity, as it probably inhibited the formation of well-organized crystalline regions and led to a more disordered, amorphous structure in the PLA matrix. In the case of EL membranes, a slight crystallinity reduction due to the presence of the porogen agent was evidenced. Indeed, X_c_ of 8-2EL was 5.8, whereas 8EL and 10EL showed values of 6.9 and 10.6, respectively. It is worth highlighting that the rise in the degree of crystallinity degree due to polymer concentration for EL membranes was more remarkable compared with that observed for Cyr, and, moreover, that 10EL exhibited the highest degree of crystallinity of all membranes. Since the PLA/EL solution was not thermodynamically favored, as explained above, once immersed in the coagulating bath, the precipitation of PLA on the top surface was immediate and created a very dense structure. In these conditions, the rate of exchange between solvent and nonsolvent was reduced and polymer chains had more time to rearrange themselves into well-defined crystalline domains during the precipitation.

Finally, as for DMI membranes, the Tg of resulting porous membranes slightly increased with PLA concentration. The other parameters were not considerably affected by the concentration increase, whereas X_c_ moderately increased with the addition of Plu.

### 3.5. Pure Water Permeability

The PWP of the obtained PLA membranes mirrors the combination of the membrane morphology and related properties (Figure 5).

PLA–Cyr membranes showed the lowest detected PWP: 10Cyr had a PWP of 10 L/m^2^hbar, followed by 8Cyr with a PWP of 13 L/m^2^ h bar, and a slight increase was observed for 8-2Cyr which had a PWP of 24 L/m^2^ h bar. The impact of the thickness was noticeable for the membranes obtained with EL, which showed higher PWP values and was also favored by the lower measured contact angle. In fact, in this case, the PWP for 10EL was 72 L/m^2^ hbar and increased to 74 L/m^2^ h bar for 8EL, until it reached 93 L/m^2^ h bar for 8-2EL. The membranes prepared with DMI as a solvent gave the most promising results in terms of PWP, leading to values higher than 300 L/m^2^ h bar for all three membranes. More precisely, 10DMI showed a PWP of 314 L/m^2^ h bar and a value that increased to 371 L/m^2^ h bar by testing the 8DMI membrane and further increased up to 389 L/m^2^ h bar by using an 8-2DMI membrane, which represented the membrane with the highest detected pore size (0.09 μm) and porosity (86.71 ± 0.35%).

It is worth mentioning that the solvent, being the component in the largest quantity during NIPS, has a significant impact on the sustainability of the membrane preparation process. Hence, the choice of green solvents such as Cyr, DMI, and EL greatly improves the environmental impact of membrane production. Moreover, the health and safety of workers improve noticeably when considering the favorable toxicological profile of the selected solvents in comparison to the commonly used toxic ones. The adopted polymer, PLA, is derived from renewable resources, so the need for fossil fuel feedstocks should be negated for the preparation of the obtained novel membranes. This study has demonstrated the feasibility of PLA-based membranes produced using biobased solvents, and although the obtained results represent a preliminary investigation, they could serve as a starting point for sustainable processes ranging from the manufacture of membranes to the separation processes in which they are employed.

## 4. Conclusions

In this work, PLA UF membranes were prepared via NIPS by using, for the first time, three different nontoxic, biobased solvents: Cyr, DMI, and EL. The composition of the casting solutions was varied in terms of polymer content as well as the presence of Plu as an amphiphilic pore former agent. Cyr allowed us to obtain PLA membranes with a pore size of 0.2–0.3 μm and a PWP in the range of 10–24 L/m^2^ h bar. When using Cyr, the final membrane properties were not strongly affected by the presence of Plu, as in the case of the other two tested biobased solvents, for which the positive effect of the amphiphilic agent was noticeable in terms of pore size and PWP increase as well as contact angle reduction. When EL was employed as a solvent, the membranes exhibited a pore size of 0.3–0.6 μm with a PWP of 72–93 L/m^2^ h bar, and further increases were registered by using DMI. In fact, PLA-DMI and PLA-PLU-DMI led to the most promising results, with membrane pore sizes of 0.02–0.09 μm and PWP in the range of 314–389 L/m^2^ h bar. All of the produced membranes showed high porosity, which was higher than 75% for all three tested solvents, and good mechanical strength of higher than 120 and reaching a maximum of 398 N/mm^2^. The detected crystallinity of PLA-EL systems was higher than that of PLA-Cyr and PLA-DMI systems; when using EL, the dense skin layer formed on the membrane surface allowed a slow crystallization under the upper surface of the membrane. The obtained data may contribute to the development of more sustainable polymeric membrane preparation based on the use of PLA as a polymeric material and solvents produced from biobased raw materials and free of toxicity and risk of carcinogenicity.

## Figures and Tables

**Figure 1 polymers-16-02024-f001:**
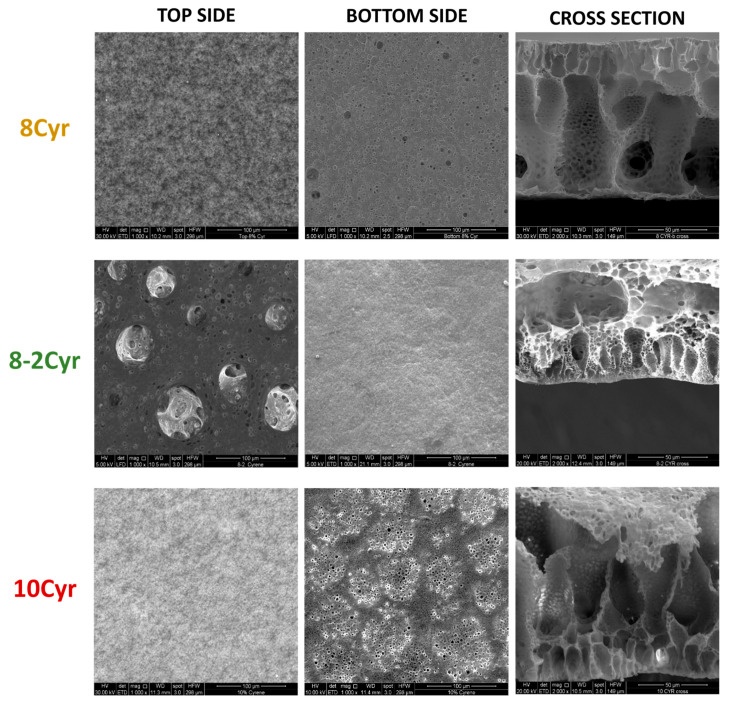
SEM micrographs of PLA membranes (top side, bottom side, and cross-section).

**Figure 2 polymers-16-02024-f002:**
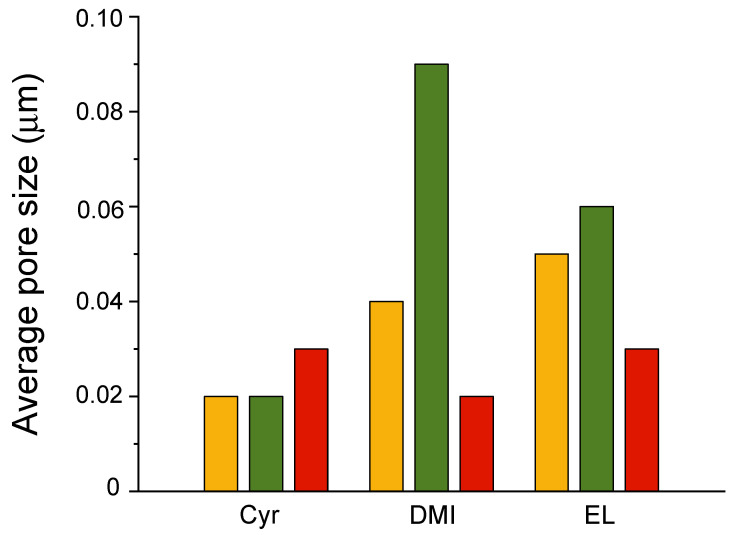
Average pore size of 8 (
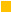
), 8-2 (
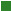
), and 10 (
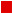
) PLA membranes.

**Figure 3 polymers-16-02024-f003:**
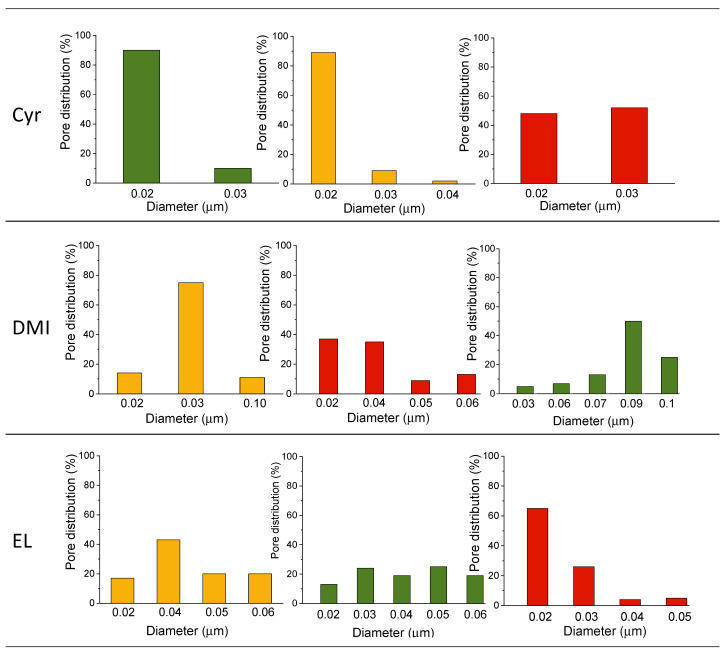
Pore size distribution (diameter vs. pore distribution %) of 8 (
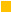
), 8-2 (
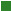
), and 10 (
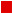
) PLA membranes.

**Figure 4 polymers-16-02024-f004:**
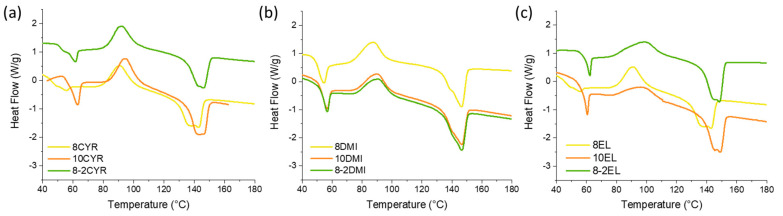
DSC thermograms of PLA membranes obtained by NIPS using Cyr (**a**), DMI (**b**), and EL (**c**) as biobased solvents.

**Figure 5 polymers-16-02024-f005:**
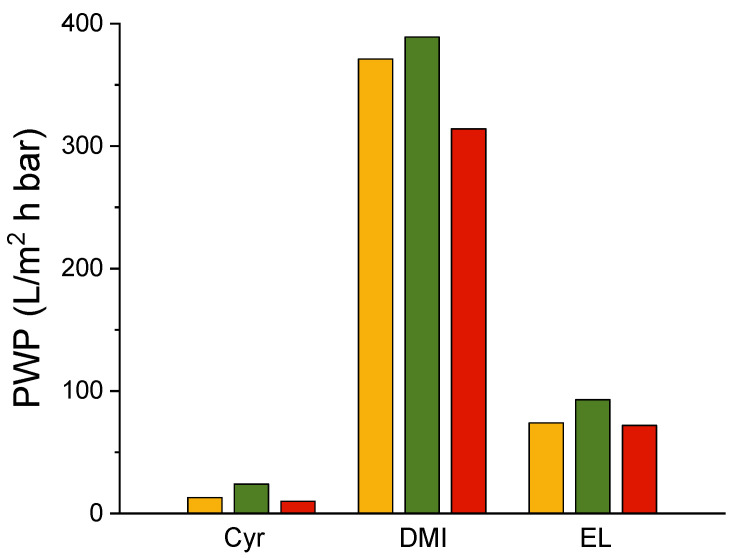
PWP of 8 (
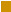
), 8-2 (
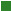
), and 10 (
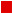
)PLA membranes.

**Table 1 polymers-16-02024-t001:** Solution composition, solvent type, solubilization temperature, and viscosity for membrane production.

Membrane Code	PLA-Plu (%)	Solvent	Dope Solution Temperature (°C)	Viscosity (cP)
8Cyr	8-0	Cyr	85	9600
8-2Cyr	8-2			14,400
10Cyr	10-0			16,800
8DMI	8-0	DMI	85	2800
8-2DMI	8-2			3200
10DMI	10-0			8000
8EL	8-0	EL	100	1200
8-2EL	8-2			1600
10EL	10-0			2000

**Table 2 polymers-16-02024-t002:** Some of the most relevant physical-chemical properties of Cyr, DMI, and EL [47,48,49].

Properties	Cyr	DMI	EL
Color	Light yellow	Colorless	Colorless
Molecular weight (g/mol)	128.13	174.2	118.13
Formula	C_6_H_8_O_3_ 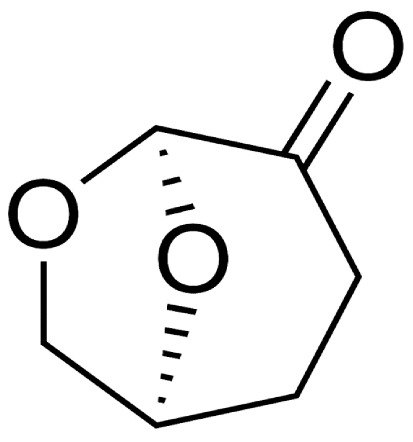	C_8_H_14_O_4_ 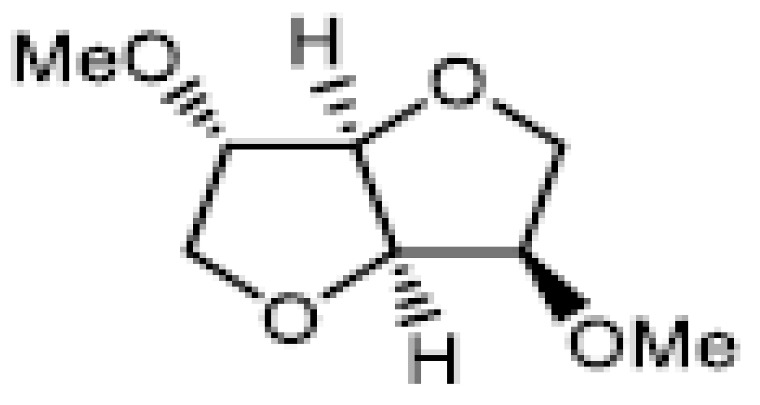	C_5_H_10_O_3_ 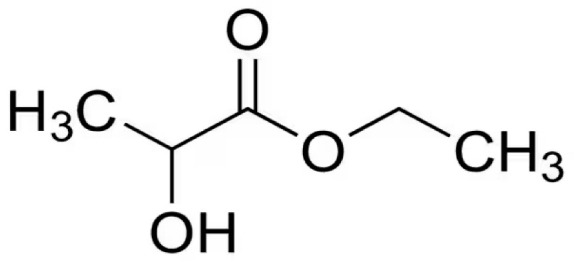
Boiling point (°C)	227	240	154
Flash point (°C-closed cup)	108	116	46
Density (g/cm^3^)	1.25	1.17	1.03
Miscibility with water (g/L at 20 °C)	Complete	2000	100
δd (MPa^0.5^)	18.8	17.6	16.0
δp (MPa^0.5^)	10.6	7.1	7.6
δh (MPa^0.5^)	6.9	7.5	12.5
PLA–solvent distance * (MPa^0.5^)	1.16	3.33	7.37
Water-solvent distance * (MPa^0.5^)	35.95	35.98	30.96

* Calculated by applying the following equation: $\Delta P_{P-S}=\sqrt{((\delta_{d,P}-\delta_{d,S}{)}^{2}+{\left(\delta_{p,P}-\delta_{p,S}\right)}^{2}+{\left(\delta_{h,P}-\delta_{h,S}\right)}^{2})}$.

**Table 3 polymers-16-02024-t003:** Thickness, porosity, and pore size of PLA membranes.

Membrane Code	Thickness (mm)	Porosity (%)
8Cyr	0.125 ± 0.007	84.59 ± 0.91
8-2Cyr	0.132 ± 0.006	86.12 ± 0.77
10Cyr	0.133 ± 0.002	83.18 ± 1.20
8DMI	0.123 ± 0.008	83.07 ± 0.44
8-2DMI	0.128 ± 0.007	86.71 ± 0.35
10DMI	0.129 ± 0.009	75.93 ± 1.57
8EL	0.090 ± 0.005	76.53 ± 1.35
8-2EL	0.087 ± 0.007	78.71 ± 2.39
10EL	0.095 ± 0.005	74.79 ± 1.22

**Table 4 polymers-16-02024-t004:** Contact angle and mechanical properties (Young’s modulus and elongation at break) of PLA membranes.

Membrane Code	Contact Angle		Mechanical Properties	
	Top Surface (°)	Bottom Surface (°)	Young’s Modulus (N/mm^2^)	Elongation at Break (%)
8Cyr	85 ± 1	88 ± 2	285 ± 7	65 ± 4
8-2Cyr	79 ± 3	90 ± 4	395 ± 3	78 ± 6
10Cyr	86 ± 2	94 ± 3	244 ± 3	63 ± 6
8DMI	81 ± 3	82 ± 3	274 ± 3	20 ± 6
8-2DMI	77 ± 1	74 ± 2	121 ± 6	18 ± 5
10DMI	89 ± 1	92 ± 2	237 ± 5	25 ± 6
8EL	81 ± 2	95 ± 1	398 ± 8	18 ± 7
8-2EL	71 ± 2	90 ± 5	347 ± 8	17 ± 3
10EL	83 ± 2	94 ± 3	275 ± 4	16 ± 3

**Table 5 polymers-16-02024-t005:** Thermal characteristics of PLA membranes prepared using the three different solvents (Cyr, DMI, and EL).

Membrane Code	T_g_ (°C)	T_cc_ (°C)	ΔH_cc_ (J/g)	T_m_ (°C)	ΔH_m_ (J/g)	Xc (%)
8Cyr	53.3	91.3	15.6	143.1	19.4	4.0
10Cyr	60.9	94.8	17.4	143.6	22.6	5.5
8-2Cyr	58.9	92.4	24.2 *	146.3	25.9 *	1.8
8DMI	51.3	87.7	16.1	146.4	23.1	7.5
10DMI	54.2	90.0	14.8	146.6	21.8	7.5
8-2DMI	55.0	90.4	18.0 *	146.5	26.9 *	9.6
8EL	60.2	100.2	20.2	145.1	26.7	6.9
10EL	61.6	97.3	12.0	149.0	21.9	10.6
8-2EL	61.4	98.6	25.4 *	148.6	30.8 *	5.8

* Referred to the actual weight fraction of PLA.

## Data Availability

Data are contained within the article.

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
