# Peer review of "Enhancing Sustainability in PLA Membrane Preparation through the Use of Biobased Solvents"

_polymers, 2024, doi:10.3390/polym16142024_

Round 1

Reviewer 1 Report

Comments and Suggestions for Authors

- Author should use precise method for porosity of membrane especially the prepared membrane is Ultrafiltration (UF). Please explain the benefit of selected method (solvent absorption).

- Equation 1 is mistake for density of solvent.

- Author should show membrane permeate flux graph and calculate for their slop to confirm the level of filtration.

- The membrane casting at 70 degreeC, the temperature of glass slide is effect on membrane morphology or not. If glass temperature effect, how can you control the membrane morphology.

- The results should change the graph type to contour for more comparative.

- Author should give more advantage or application of prepared membrane.  

Comments on the Quality of English Language

This manuscript is very interested with biocompatible membrane but author should improve the language to avoid repeat sentence.

Reviewer 2 Report

Comments and Suggestions for Authors

The manuscript needs minor revision

1) Figure 1 shows that the morphology of the membrane's top and bottom surfaces is very different. The reviewer wonders if PWP is different when water is fed from the sides of the top and bottom surfaces. Please comment.

2) The ultrafiltration membrane is used for separation. It may not be sufficient to only test PWP. Please comment.

Round 2

Reviewer 1 Report

Comments and Suggestions for Authors

The manuscript was revised as comments.

Comments on the Quality of English Language

The manuscript should be revised by native speaker for language writing.